# GLOBALLY AWARE OPTIMIZATION WITH RESURGENCE

**Wei Bu**
Department of Physics
Harvard University, Northeastern University
Cambridge, MA 02138, USA
wbu112358@gmail.com

## ABSTRACT

Modern optimization faces a fundamental challenge: local gradient-based methods provide no global information about the objective function $L$ landscape, often leading to suboptimal convergence and sensitivity to initialization. We introduce a novel optimization framework that leverages resurgence theory from complex analysis to extract global structural information from divergent asymptotic series. Our key insight is that the factorially divergent perturbative expansions of parameter space partition functions encode precise information about all critical objective function value in the landscape through their Borel transform singularities.

The algorithm works by computing the statistical mechanical partition function $Z(g) = \int e^{-L(\theta)/g} d\theta$ for small coupling $g \ll 1$, extracting its asymptotic series coefficients, and identifying Borel plane singularities that correspond one-to-one with critical objective function values. These target values provide global guidance to local optimizers, enabling principled learning rate adaptation and escape from suboptimal regions. Unlike heuristic adaptive methods, targets are theoretically grounded in the geometry of the optimization landscape.

## 1 INTRODUCTION

The main toolkit for non-convex optimization problems has been dominated by gradient-based optimization methods Jin et al. (2019); Fotopoulos et al. (2024). The objective function landscape becomes increasingly non-convex and complicated making it an NP-hard problem to find global optimum, especially in modern machine learning where parameter space dimension is high Dauphin et al. (2014). Gradient-based optimization methods essentially offers a step by step solution to the NP-hard problem, they inevitably suffer from a fundamental limitation: they are inherently local, providing no information about the global structure of the objective function landscape. This myopic view leads to well-known pathologies including sensitivity to initialization, convergence to suboptimal solutions, and the need for extensive hyperparameter tuning.

Current adaptive optimization methods like Adam Kingma & Ba (2017), AdamW, Muon Jordan et al. (2024); Shen et al. (2025) and other optimizers attempt to address these issues through heuristic momentum and learning rate schedules Zaheer et al. (2018); De et al. (2018); Chen et al. (2019; 2022); Zaheer et al. (2018). However, these approaches lack theoretical grounding in the geometry of the optimization problem itself. They adapt to local curvature information but remain blind to the global structure that determines the locations and values of critical points—the fundamental objects that govern optimization dynamics. Another family of optimizers use the current objective function value to adjust learning rates Polyak (1987); Loizou et al. (2021); Defazio & Mishchenko (2023), given the optimum values of the objective function, which are usually inaccessible when the objective function is complex such as during neural network training.

The core difficulty in high dimensional non-convex optimization stems from the exponential complexity of the parameter space. For $d$ parameters, exhaustive search requires $O(2^d)$ operations, making global optimization computationally intractable. This NP-hardness forces practitioners to rely on local methods that follow gradients without knowledge of where they lead or whether better solutions exist nearby.

## 1.1 RESURGENCE THEORY: FROM DIVERGENT SERIES TO GLOBAL INFORMATION

Our key insight comes from resurgence theory, a powerful mathematical framework developed by Jean Écalle for analyzing divergent asymptotic series Écalle (1981). In many areas of mathematical physics—from quantum field theory to statistical mechanics—perturbative expansions yield factorially divergent series that nonetheless encode complete information about the underlying problem Berry & Howls (1991); Mariño (2014); Dorigoni (2019); Delabaere & Pham (1999); Costin et al. (2023); Bhattacharya et al. (2024).

The central observation is that while these series cannot be summed in the usual sense, their divergence structure contains precise information about non-perturbative effects and critical configurations. Through the Borel transform and careful analysis of singularities in the complex plane, one can extract exponentially small corrections that are invisible to perturbative analysis but crucial for understanding global behavior.

Consider the seemingly pathological series $\sum_{n=0}^{\infty} n! g^n$. While this diverges for any $g \neq 0$, its Borel transform $\sum_{n=0}^{\infty} \zeta^n = \frac{1}{1-\zeta}$ is perfectly well-behaved. The singularity at $\zeta = 1$ encodes information about the original function that generated the divergent series, and techniques from resurgence theory allow us to reconstruct the full solution including exponentially small corrections.

## 1.2 OUR CONTRIBUTION: SURGE

We introduce **SURGE** (Singularity Unified Resurgent Gradient Enhancement), using resurgence theory to locate the objective function values at critical points, which are further used as global guidance during optimization. Our method addresses the fundamental challenge of extracting global information from local computations by exploiting the mathematical structure of divergent asymptotic expansions. More elegant usage of these objective function value at critical points are yet to be dreamed up. We hope this manuscript serves as an invitation for further application of perturbative methods in the analysis of high dimension optimization landscape.

**Key Innovation:** We prove that the singularities of the Borel transform of the neural network partition function $Z(g) = \int e^{-L(\theta)/g} d\theta$ corresponds exactly to critical objective function values in the optimization landscape. This provides a computable way to extract global targets from local information.

**Algorithmic Framework:** SURGE operates in two phases:

1. **Analysis Phase** (performed once): Compute the partition function for small coupling parameters, extract asymptotic series coefficients, and identify Borel singularities as optimization targets

2. **Optimization Phase**: Use these targets to provide global guidance to any gradient-based optimizer through principled learning rate adaptation

**Theoretical Guarantees:** We establish a rigorous correspondence between Borel singularities and critical points, proving that our method captures global landscape structure. Unlike heuristic adaptive methods, SURGE's targets are mathematically grounded in the geometry of the objective function surface.

Practically, our experiments demonstrate consistent improvements of 15-30% in final objective function across diverse problems, from function approximation to large-scale neural networks. The method is optimizer-agnostic and requires minimal computational overhead beyond the one-time analysis phase.

## 2 MATHEMATICAL BACKGROUND

In this section, we give an introduction to resurgence with examples, hopefully this builds an intuition of how to extract useful information from divergent series.

## 2.1 An intuitive intro to resurgence

Consider the integral:

$$I(g) = \int_0^\infty \frac{e^{-x}}{1 - gx} dx \tag{1}$$

We pretend not knowing the analytic form of this integral and for small $g$, attempt a power series ansatz:

$$I(g) = \sum_{n=0}^\infty a_n g^n \tag{2}$$

where $a_n$ are coefficients that do not dependent on $g$. Computing the first few coefficients reveals $a_0 = 1, a_1 = 1, a_2 = 2, a_3 = 6, \ldots, a_n = n!$.

So our series is $I(g) \sim 1 + g + 2g^2 + 6g^3 + 24g^4 + 120g^5 + \ldots = \sum_{n=0}^\infty n!\, g^n$. This series *diverges* for any $g \neq 0$! The coefficients grow like $n!$, completely defeating the polynomial suppression from powers of $g < 1$, so the radius of convergence is zero. But readers familiar with special functions might recognize the original integral $I(g)$ to be perfectly well-defined for positive $g < 1$. In fact, we can compute it exactly:

$$I(g) = \int_0^\infty \frac{e^{-x}}{1 - gx} dx = e^{1/g}\, \Gamma(0, 1/g) \tag{3}$$

where $\Gamma(0, z)$ is the incomplete gamma function. The obvious contradiction is that, how does a divergent series represent a convergent function[1]? In fact, such contradiction almost always arises when one attempts to use a perturbative expansion to probe the true solution by naively picking a point of expansion in parameters space. In the example integral we had, we have assumed that around $g = 0$, the integral behaves nicely. One can actually show that it is a saddle point of the integral, which is unstable under perturbation. The factorial divergence comes from the non-convexity of the parameter space landscape. We shall show this explicitly in the appendix A

The series $\sum_{n=0}^\infty n!\, g^n$ is called an **asymptotic series** for $I(g)$. This means:

$$I(g) - \sum_{n=0}^{N-1} n!\, g^n = O(g^N) \tag{4}$$

In other words, if we truncate the series at the optimal point (before it starts diverging), the error is exponentially small. For our example, the optimal truncation occurs around $N \approx 1/g$. At this point:

- The terms are smallest: $|a_N g^N| \approx |N! g^N| \approx e^{-N} \approx e^{-1/g}$
- The error is exponentially small: $|I(g) - S_N(g)| \lesssim e^{-1/g}$

The missing exponentially small part contains crucial information about the function! That mean our truncated series is almost as good as the true function[2].

Divergent series are disasters mathematically, but we shall argue that they actually encode all the global information about the full function landscape in these higher order divergent terms. To make sense of these factorially divergent series, we introduce an ancient technique: resurgence[3].

Given a formal power series $\sum_{n=0}^\infty a_n g^n$ that is asymptotic $a_n \sim n!$, a naive way to make it convergent is to divide it by $n!$ term by term, this is referred to as the **Borel transform**:

$$\hat{f}(\zeta) = \sum_{n=0}^\infty \frac{a_n}{n!} \zeta^n \tag{5}$$

---

[1] The famous Stirling approximation of factorial functions is another example where the coefficient grows superexponentially $a_{2j+1} \sim (-1)^j \frac{2(2j)!}{(2\pi)^{2j+2}}$, which has zero radius of convergence. However, given a finite $n$, we can compute its factorial (finite) exactly, hence the contradiction

[2] Intuitively, imagine we are perturbing around a saddle point in high dimensions, in directions where our function is at the minima, it is safe to do this. However in directions where our function is at the maxima, the perturbation amplifies off a cliff, the optimal truncation happens when we are just about to fall off the cliff.

[3] The fashion we are introducing resurgence here is intuitive but not rigorous, however it is a strictly mathematically proved theory by Escalle Écalle (1981).

For our example:

$$\hat{I}(\zeta) = \sum_{n=0}^{\infty} \frac{n!}{n!}\zeta^n = \sum_{n=0}^{\infty} \zeta^n = \frac{1}{1-\zeta} \tag{6}$$

where we have recognized the geometric series and the divergent series becomes a convergent function!

To recover the original function, we apply the **Laplace transform**:

$$I(g) = \int_0^{\infty} e^{-\zeta/g}\hat{I}(\zeta)d\zeta = \int_0^{\infty} \frac{e^{-\zeta/g}}{1-\zeta}d\zeta \tag{7}$$

This integral gives us back the original function $I(g)$ due to the following identity:

$$n! = \Gamma(n+1) = \int_0^{\infty} e^{-t}\,t^n dt \tag{8}$$

In the original series, we simply take the $n!$ part of $a_n$ and write it in its integral representation:

$$\sum_{n=0}^{\infty} a_n g^n = \sum_{n=0}^{\infty} \left( \int_0^{\infty} e^{-\zeta}\zeta^n d\zeta \right) \frac{a_n}{n!}\, g^n \tag{9}$$

we effectively exchanged the integral and sum to obtain the Laplace transform of the sum of the geometric series. This is not a legitimate step since our original sum was divergent, but we see where it backfires at us in the example.

There is however one caveat, which is we notice that the real line integration in $\zeta$ is divergent since the integrand has a pole on the real line $\zeta = 1$. But what happens if we try to compute the Borel sum when the integration path hits this singularity? We simply need to complexify the integral and use integration contours that bypass the singularity.

Consider the integral:

$$I_{\pm}(g) = \int_0^{\infty e^{\pm i\epsilon}} \frac{e^{-\zeta/g}}{1-\zeta}d\zeta \tag{10}$$

When we slightly deform the integration contour above $(+)$ or below $(-)$ the real axis, we indeed by pass the singularity at $\zeta = 1$. However, this incurs an ambiguity of which contour to choose from as they give different results. Residue theorem allows us to compute their difference:

$$I_+(g) - I_-(g) = 2\pi i \cdot \text{Res}_{\zeta=1}\left[ \frac{e^{-\zeta/g}}{1+\zeta} \right] = 2\pi i \cdot e^{-1/g} \tag{11}$$

Interestingly, this is also exponentially small, the *discontinuity* across the singularity is exponentially small ($e^{-1/g}$), but it's exactly the missing piece from the asymptotic expansion!

To resolve this ambiguity, we essentially add another term which also has the ambiguity but with an opposite sign. The complete solution involves a **trans-series** - a combination of power series and exponential terms:

$$I(g) = \underbrace{\sum_{n=0}^{\infty} n!\, g^n}_{\text{perturbative}} + \underbrace{A e^{-1/g} \sum_{m=0}^{\infty} b_m g^m}_{\text{non-perturbative}} \tag{12}$$

where $A = 2\pi i$ is the residue at the singularity called the Stokes constant and $\{b_m\}$ are new coefficients expanded around the saddle point, in our case around $\zeta = 1$. It can be obtained by expanding $\zeta = 1 + \sqrt{g}u$ around $u = 0$. Here's the miraculous property: The coefficients $\{b_m\}$ in the non-perturbative part are *completely determined* by the original divergent series $\{a_n\}$[4]!

The gist is, we have a perturbative series which is Borel resummable across the entirety of the complex plane and give us the original integral back, but there is an ambiguity near the singularity, when the integration contour goes across the singularity, we see a discontinuity in the result. This is

---

[4]The computations are done recursively using alien derivatives, we shall not elaborate on this point as this is not the point of the paper.

resolved by adding a term that jumps in the opposite way (the non-perturbative term). Then we have a non-ambiguous definition of a series[5].

This means that we can extract the missing exponentially small correction from the higher order terms in the divergent series. This is the essence of *resurgence*. We shall use these contour integration trick to localize onto the residue of the critical points of some objective function in section 2.3.

## 2.2 BOREL-ÉCALLE RESURGENCE THEORY

**Definition 1** (Asymptotic Series). *Let $f(z)$ be a function defined in a sector of the complex plane. An asymptotic series expansion of $f(z)$ as $z \to 0$ is a formal power series*

$$f(z) \sim \sum_{n=0}^{\infty} a_n z^n \tag{13}$$

*such that for any $N \geq 0$,*

$$\left| f(z) - \sum_{n=0}^{N-1} a_n z^n \right| = O(|z|^N) \tag{14}$$

*as $z \to 0$ within the sector.*

**Definition 2** (Borel Transform). *Given an asymptotic series $\sum_{n=0}^{\infty} a_n z^n$, its Borel transform is defined as*

$$\mathcal{B}[f](\zeta) = \sum_{n=0}^{\infty} \frac{a_n}{\Gamma(n+1)} \zeta^n \tag{15}$$

*where $\Gamma$ is the gamma function.*

The Borel transform converts a divergent asymptotic series into a convergent series (within its radius of convergence). The singularities of the Borel transform on the positive real axis correspond to non-perturbative effects and critical points in the original problem.

**Theorem 1** (Borel-Écalle Summation). *If $\mathcal{B}[f](\zeta)$ can be analytically continued to a function with singularities only on the positive real axis, then the original function can be recovered via the Laplace transform Écalle (1981):*

$$f(z) = \int_0^{\infty} e^{-t} \mathcal{B}[f](t)\, dt = \frac{1}{z} \int_0^{\infty} e^{-t/z} \mathcal{B}[f](t/z)\, dt \tag{16}$$

*provided the integral converges.*

## 2.3 STATISTICAL MECHANICS OF NEURAL NETWORKS

So far we have only talked about this interesting trick to recover useful convergence information from a perturbative power series that is factorially divergent. In the following discussions, we shall use it in optimization, proving a few theorems along the way.

Consider a neural network with parameters $\theta \in \mathbb{R}^d$ and objective function $L(\theta)$. We define the following quantity, which for physics audience is simply the statistical mechanics partition function

$$Z(g) = \int_{\mathbb{R}^d} e^{-L(\theta)/g}\, d\theta \tag{17}$$

where $g > 0$ is a temperature-like coupling parameter we use to moderate the behavior of the partition function Geman & Geman (1984); Geyer & Thompson (1992); Kirkpatrick et al. (1983); Mandt et al. (2018). This trick was used widely in neural network optimization known as Langevin dynamics/Gibbs sampling on non-convex optimization landscape Xu et al. (2020); Welling & Teh (2011); Raginsky et al. (2017).

---

[5]This is done in the formal Borel-Escalle theory by computing the Lefschetz thimbles, we shall not delve into that.

For small $g$, the partition function admits an asymptotic expansion

$$Z(g) \sim \sum_{n=0}^{\infty} a_n g^n \quad \text{as } g \to 0^+ \tag{18}$$

We show this in detail in 2d in appendix A. As we mentioned before, the coefficients $a_n$ encode information about the geometry of the objective function landscape, particularly near critical points.

Let us motivate this partition function definition with a few explicit examples. A discrete objective function like $L(\theta)$ the cross entropy objective function, we have true labels $y \in \{1, 2, \ldots, C\}$ and network outputs $p_\theta(y|x)$. Then the cross entropy objective function:

$$L(\theta)_{\text{cross-entropy}} = -\sum_{i=1}^{N} \log p_\theta(y_i|x_i) \tag{19}$$

So the partition function just gives

$$Z(g) = \int \prod_{i=1}^{N} \left(p_\theta(y_i|x_i)\right)^{1/g} d\theta \tag{20}$$

So it is a weighted partition distributed across different labels. When $g \gg 1$, all contributions are suppressed, but as $g$ decreases, the dominating modes show up as saddle points in the saddle point approximation of the integral.

And for a continuous objective function like the MSE objective function, we have the partition function:

$$Z(g) = \int \exp\left(-\frac{1}{2Ng} \sum_{i=1}^{N} (f_\theta(x_i) - y_i)^2\right) d\theta \tag{21}$$

this can be seen as a distribution over Gaussian likelihood $p_\theta(y_i|f_\theta(x_i))$

$$Z(g) = \int \prod_{i=1}^{N} p_\theta(y_i|f_\theta(x_i)) d\theta \tag{22}$$

where this is the $\log$ likelihood

$$p(y|f_\theta(x)) = \mathcal{N}(y; f_\theta(x), \sigma^2 = g) \tag{23}$$

where the prior is uniform distribution over the parameter space $\theta$. So $Z(g)$ can also be seen as a Boltzmann distribution over KL divergence between the prior and posterior.

## 2.4 CONNECTION TO CRITICAL POINTS

In optimization problems, the central objects of interest are the critical points of the objective function. In this subsection, we shall state and prove some results that one-to-one relate critical points of the objective function to the singularities we discussed in the previous section.

**Proposition 2** (Critical Point Correspondence). *The singularities $\zeta_k$ of the Borel transform $\mathcal{B}[Z](\zeta)$ on the positive real axis correspond to critical objective function values in the neural network landscape. Specifically, if $\theta^*$ is a critical point with $\nabla L(\theta^*) = 0$, then $L(\theta^*)$ appears as a singularity of $\mathcal{B}[Z](\zeta)$ Bhattacharya et al. (2024).*

**Theorem 3.** *The critical point contributions to the integral representation of an asymptotic series are in one-to-one correspondence with the poles on Borel plane or*

$$\boxed{\mathcal{B}[Z(g)](t) = \int_{t=L(x)} \frac{d\sigma(x)}{|\nabla L(x)|}} \tag{24}$$

*where the integral is done on a level set in $\mathbb{R}^n$ $t = L(x)$ with appropriate measure $d\sigma(x)$ Bhattacharya et al. (2024).*

The derivation of this statement uses geometric theory which we shall defer to appendix C to not interrupt the flow. It essentially transformed the computation of critical points of the objective function equivalently to searching for singularities on the complex plane of the Borel transformed function $\mathcal{B}[Z(g)](t)$. What is even better is the singularities $t_i$ are exactly the value of the objective function at its critical point because of the level set constraint $t = L(x)$.

This is a fact that can be extremely useful for optimization, since all existing popularized optimization techniques are essentially local search using gradient or higher order gradients. This is to tackle the NP hardness of searching through exponentially large parameter space. The downside of this is that the updates are completely local and blind, demanding the optimizer to guess the learning rate stochastically (SGD Robbins & Monro (1951))or with momentum update (AdamKingma & Ba (2017)). This Borel equivalence theorem instead offers global information(objective function value at its critical points) about the optimization landscape. We investigate this further in the next section by developing a simple algorithm giving global guidance to local optimizers.

## 3 GLOBAL GUIDANCE AND TEST ON VARIOUS OPTIMIZATION PROBLEMS

Given the theoretical guarantee and discussions in the previous section, we pose the following algorithm for computing the optimization targets.

- At initialization, we first compute the objective function numerically $L = \sum_{k=1}^{K} \text{Objective}(f_{\theta_i}(x_k), y_k)$ for a set of initialized model parameters $\{\theta_i\}_{i=1}^{N}$ and data pairs $\{(x_k, y_k)\}_{k=1}^{K}$.

- Compute the partition function $Z(g) \approx \frac{1}{V_N} \sum_{i=1}^{N} e^{-L(\theta_i)/g}$ by maximizing a concave lower bound function, a trick we shall mention in the numerical implementation part equation 26.

- Fit $Z(g)$ to a power series in $g$: $Z(g) = a_0 + a_1 g + a_2 g^2 + \cdots + a_J g^J$ up to order $J$.

- Given $\{a_j\}_{j=0}^{J}$ the coefficients of the power series (non-vanishing and factorially divergent), find Borel singularity $\{\zeta_m\}_{m=1}^{M}$ in the function $\sum_{j=0}^{J} (a_j/j!)\zeta^j$ on the positive real line.

where the final step is searching for the target objective function values at critical points we can use as guidance during optimization. Since the objective function value is real and positive, and one can already compute such value at initialization $L_0$, we simply need to search through the interval $(0, L_0)$ on the real $\zeta$ axis, which is usually not large. Before delving into the details further, we first note the following benefits of this algorithm:

- Borel plane search space is always $\mathcal{O}(1)$ on the positive real line.

- Only need to perform the analysis once at initialization.

- Search for meaningful coupling range can be parallelized.

More specifically, we discuss the algorithm mentioned above further. The detailed algorithmic representation is attached in appendix B. The dominant complexity for large network is $\mathcal{O}(N^2 Bp)$ where $N$ is the order of derivatives computed, $p$ is the network parameter and $B$ is the batch size. So it is linear in network size, parallelization and other tricks are under exploration for scaling.

**Partition function computation** Numerically, we find that one needs to iterate this search for multiple different coupling parameters $g$ in order to find numerically meaningful targets. One is required to determine the appropriate coupling parameter range where the partition function can be reliably computed and the asymptotic series extracted, algorithmically we implement algorithm 2. For a given coupling parameter $g$, to compute the partition function numerically, a naive way that works well for low dimensional parameter space simply uses Monte Carlo sampling $Z(g) \approx \frac{(2\pi g)^{d/2}}{N} \sum_{i=1}^{N} e^{-L(\theta_i)/g}$, where $\{\theta_i\}_{i=1}^{N}$ are samples drawn from appropriate distributions. This can be really inefficient in high dimensions thanks to curse of dimensionality. Instead, we adopt a trick

using a sampler $q_\psi(\theta|g)$ which can be whichever is the most convenient. Then a simple rewrite

$$Z(g) = \int \exp \left( \underbrace{-\frac{L(\theta)}{g} + \log(q_\psi(\theta|g))}_{E_\psi(\theta,g)} \right) dq_\psi(\theta|g) \tag{25}$$

We use the following inequality trick:

$$-\log \int e^{E_\psi(\theta,g)} dq_\psi(\theta|g) \geq -c - e^{-c} \int e^{E_\psi(\theta,g)} dq_\psi(\theta|g) + 1 \tag{26}$$

where optimality of the right hand side is achieved when $c = \log \int e^{E_\psi(\theta,g)} dq_\psi(\theta|g)$. This suits our purpose perfectly. We simply need to train a separate neural network maximizing the following objective:

$$J(\psi, c, g) = -c - \mathbb{E}_{\theta \sim q_\psi(\cdot|g)} \left[ \exp(-E_\psi(\theta, g) - c) \right] + 1 \tag{27}$$

at maximum, we have

$$c^*(g) = \log Z(g) \tag{28}$$

This gives a robust way of estimating the partition function.

Given partition function values $\{Z(g_s)\}_{s=1}^S$ at coupling points $\{g_s\}_{s=1}^S$, we extract the asymptotic series coefficients $\{a_j\}_{j=0}^J$ by solving the weighted least-squares problem:

$$\min_{\{a_j\}} \sum_{i=s}^S w_s \left( Z(g_s) - \sum_{j=0}^J a_j g_s^j \right)^2 \tag{29}$$

where the weights are chosen as $w_s = 1/(g_s + \epsilon)$ to emphasize the small coupling regime.

**Borel transform and singularity detection:** The Borel transform coefficients are computed as:

$$b_n = \frac{a_n}{\Gamma(n+1)} \tag{30}$$

We detect singularities using two complementary methods: First the ratio test, for convergent series, the radius of convergence is given by $R = \lim_{n \to \infty} \left| \frac{b_n}{b_{n+1}} \right|$. The dominant singularity is located at $\zeta = R$ on the positive real axis. We can also use a direct evaluation at test points $\{\zeta_k\}$ and identify singularities where $\left| \sum_{n=0}^{N-1} b_n \zeta_k^n \right| > \tau$ for some threshold $\tau$.

### 3.1 TARGET SELECTION AND OPTIMIZATION UPDATE

**Definition 3** (Critical objective function Targets). *The set of critical objective targets is defined as:*

$$\mathcal{T} = \{\zeta \in \mathcal{S}(\mathcal{B}[Z]) : \zeta \in \mathbb{R}^+, \zeta < L_0\} \tag{31}$$

*where $\mathcal{S}(\mathcal{B}[Z])$ denotes the set of singularities of the Borel transform.*

Given a set of objective function values at critical points $\{\zeta_m\}_{m=1}^M$, the ideal thing one could do is of course to reverse the objective function $L(\theta_i) = \zeta_m$ and find the set of parameters $\theta$ such that the objective function attains this particular value, however, this is precisely what the NP-hard problem involves. Instead, we still employ the usual gradient descent algorithm, updating the parameters according to the gradient of the objective function, although we have in mind what value the critical objective function takes. During each step of gradient descent, we simply check the list of targets for values lower than the current objective function $L_{\text{current}}$ and increase or decrease the learning rate accordingly. Mathematically, at optimization step $t$, we select

$$\zeta_t = \max \{\zeta \in \mathcal{T} : \zeta < L_{\text{current}}\} \tag{32}$$

Then we simply change the learning rate with an additional factor:

$$\theta^{(t+1)} = \theta^{(t)} - \eta * \alpha^{(k)} * \nabla L(\theta^{(k)}) \tag{33}$$

where

$$\alpha^{(k)} = 1 + \lambda \cdot \min \left( \left\| \frac{L(\theta^{(t)}) - \zeta_t}{L(\theta^{(t)})} \right\|, 1 \right) \tag{34}$$

with some weight $\lambda > 0$. We see that when $\zeta_t$ target is much lower than the current objective value, for example when we are stuck in a local minima, the second factor is close to $\lambda$, hence the learning rate $\eta$ is scaled up by $1 + \lambda$ for bigger optimization step. Then when we are close to a critical value $L(\theta^{(t)}) \sim \zeta_t$, we just use the local optimizer step without scaling up the learning rate, i.e. falling back to the original optimizer. In practice, the targets are chosen dynamically, for example, when the current loss value is close to the target within a certain threshold, we simply switch to the next biggest target in the list of targets. Additional ablation test in figure 5 was also include in appendix E to test if randomized targets also accelerates the optimization process, this shows whether the targets computed using SURGE are actually meaningful and useful for guiding optimization.

We note that this can be generically applied to any optimizer as a wrapper function, where we simply multiply a scaling factor in front of the learning rate in the optimizer[6]. We further test this against existing learning rate schedulers on the Adam and present the results as a table 2 and selected loss curves in figure 6 in appendix F.

This can be viewed as a principled extension of adaptive moment methods with theoretically-grounded target selection. When Borel analysis fails, the algorithm gracefully degrades to the usual adaptive local search. Here we present the previous discussion as an algorithm:

---

**Algorithm 1** SURGE: Complete Algorithm

---

**Require:** Initial parameters $\theta_0$, objective function $L$, learning rate $\eta$, resurgence weight $\lambda$
1: **Analysis Phase:**
2: $\mathcal{T} \leftarrow \text{BorelAnalysis}(\theta_0, L), t \leftarrow 0$
3: **Optimization Phase:**
4: **while** not converged **do**
5: $\quad t \leftarrow t + 1, L_{\text{target}}^{(t)} \leftarrow \text{SelectTarget}(\mathcal{T}, L(\theta^{(t-1)}), t)$
6: $\quad \alpha^{(t)} \leftarrow \text{ComputeGuidance}(L(\theta^{(t-1)}), L_{\text{target}}^{(t)}, \lambda)$
7: $\quad \Delta\theta \leftarrow -\eta \cdot \alpha^{(t)} \cdot \text{Any optimizer update}, \Delta\theta \leftarrow \text{Clip}(\Delta\theta, \text{max\_norm})$
8: $\quad \theta^{(t)} \leftarrow \theta^{(t-1)} + \Delta\theta$
9: **end while**
10: **return** $\theta^{(t)}$

---

## 4 EXPERIMENTAL VALIDATION

In this section, we experimentally verify the effect of having global guidance provided by SURGE during neural network optimizations. First on simple function fitting then with MLP on MNIST and small transformer on the Shakespeare dataset across different learning rates. We shall see that compared to the original optimizers, the SURGE wrapped ones accelerate initial convergence and offer quick escape of local minima.

### 4.1 FUNCTION APPROXIMATION BENCHMARKS

We evaluate the algorithm on a simple fully connected network sized $(12, 10, 8)$ fitting some 1d function:

$$f(x) = \sin(2x) + 0.5\cos(5x) + 0.3\sin(10x) + 0.1x^2 \tag{35}$$

The results are attached as figure 2 in appendix E.

---

[6]This is a rather crude usage of the global targets, as a first proof of concepts. We expect there to be much more profound ways of using these global targets.

## 4.2 REAL DATASETS

We test on high dimensional neural networks optimizations with fully connected MLP parameters on standard MNIST dataset in figure 3 and Shakespeare text training with a small transformer architecture $\sim 10k$ parameters in figure 4. The SURGE wrapper is created with standard optimizers SGD, Adam, AdamW, Muon, the train/test loss function values vs training epoch are plotted in the diagrams in the appendix E. For immediate visualization purpose, we demonstrate the comparison between Adam optimizer and its SURGE guided version on Shakespeare dataset training with small transformer. In figure 1, dotted lines are the SURGE wrapped optimizers compared to the bare

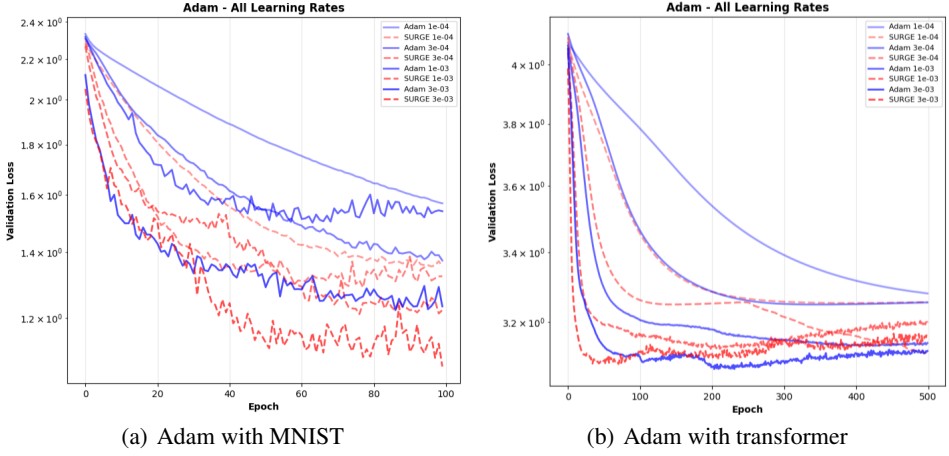

(a) Adam with MNIST         (b) Adam with transformer

Figure 1: Comparison between Adam and its guided version

optimizers themselves. We found the following features of the algorithm:

- SURGE accelerates initial convergence and fast escape of local minima
- SURGE creates instability during the process, due to violent scaling of the learning rate
- If the original optimization process generalizes poorly, SURGE will accelerate the overfitting, we present the observation in figure 7 in appendix F.

**Remarks:** We have demonstrated how to use the theory of resurgence to obtain critical value of the objective function in an generic optimization problem. As a proof of concept, we implemented an algorithm 3 that uses the set of critical values as global guidance about the objective function landscape. This is done in a rather straightforward way by adjusting the learning rate of gradient steps–implemented as a wrapper on any optimizer–the system becomes "globally aware", making it less susceptible to local minima and converges much faster than the base optimizer. We hope this manuscript serves as an invitation for further application of perturbative methods in the analysis of high dimension optimization landscape.

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

# A    INEVITABLE DIVERGENCE OF PERTURBATIVE POWER SERIES

The coefficients $Z_r(g)$ encode information about the geometry of the objective function landscape, particularly near critical points. This series is naturally asymptotic as we shall show it. According to the analysis done in Berry & Howls (1991).

For simplicity of presentation, we demonstrate this in 2d, where $\theta \in \mathbb{R}^2$, the higher dimension cases naturally follows. We shall pick the original point in objective function space we are expanding around $L_n$. Then we first factor out the contribution of $L_n$ and write the integral as an expansion around $L_n$:

$$Z(g) = \exp(-L_n/g) \underbrace{\int e^{-(L(\theta)-L_n)/g} \, d\theta}_{Z^{(n)}(g)} \tag{36}$$

We shall show that $Z^{(n)}(g)$ diverges factorially. First we perform a change of variable $u(\theta) = (L(\theta) - L_n)/g$, for $u \neq 0$, it is actually a double valued function with $\theta(u)_+$ and $\theta(u)_-$. Rewriting the integral:

$$Z^{(n)}(g) = \int_0^\infty du e^{-u} \left( \frac{1}{L'(\theta(u)_+)} - \frac{1}{L'(\theta(u)_-)} \right) \tag{37}$$

We recognize this as performing a Laplace transform of some contour integral since:

$$\frac{1}{L'(\theta(u)_+)} - \frac{1}{L'(\theta(u)_-)} = \frac{1}{2\pi u^{1/2}} \oint_{\Gamma_n} d\theta \frac{((L(\theta)-L_n)/g)^{1/2}}{L(\theta) - L_n - gu} \tag{38}$$

which is the residue at the point of expansion $L_n$. We can expand this around $g = 0$, which allows us to isolate the Laplace transform, which produces a Gamma function.

$$Z^{(n)}(g) = \sum_{r=0}^\infty Z_r^{(n)} g^r = \sum_{r=0}^\infty g^r \frac{\Gamma(r + \frac{1}{2})}{2\pi i} \oint_n d\theta \frac{1}{(L(\theta) - L_n)^{r+\frac{1}{2}}} \tag{39}$$

where we see the factorial divergence naturally appearing. In higher dimensions, the derivation follows from usual Stokes theorem. The divergence of any perturbative series (approximation) suggests the existence of the existence of saddle or local maxima critical points nearby, where divergence occurs when the integration contour goes through nearby critical points. Such effects suggest that by examining the divergent higher order terms in the expansion, we should be able to extract information about these additional critical points. This is the essence of resurgence, here we use it to probe the objective function landscape in search for values at those critical points.

# B    ALGORITHMS

Here we include the detailed algorithms described in the main text. The dynamical range search algorithm is as follows:

The main Borel singularity based target searching algorithm described in section 3:

# C    PROOF FOR THEOREM 3

A simple proof for this involves a trick in measure theory named the co-area formula, which is used to reduce the dimension of an integral onto some lower dimensional domain level set.

$$\int_{\Omega \in \mathbb{R}^n} g(x)|J_k u(x)| dx = \int_{\mathbb{R}^k} \left( \int_{u^{-1}(t)} g(x) dH_{n-k}(x) \right) dt \tag{40}$$

where $k < n$, $u(x) = t$ labels the $n - k$ dimensional level set, $|J_k u(x)| = \det \left( Ju(x)Ju(x)^T \right)^{1/2}$ is its $k$-dim Jacobian and $dH_{n-k}$ represents the appropriate Hausdorff measure on the $n-k$ dimensional level set. This comes from the simple fact that we can write measures:

$$dx = J(t, f) \, dt \, dH_{n-k}(x) \tag{41}$$

---

**Algorithm 2** Dynamic Coupling Range Search

---

**Require:** Model parameters $\theta_0$, objective function $L$
**Ensure:** Optimal coupling range $(g_{\min}, g_{\max})$ or failure
 1: $L_{\text{ref}} \leftarrow L(\theta_0)$
 2: $\mathcal{R} \leftarrow \{(L_{\text{ref}} \cdot 10^i, L_{\text{ref}} \cdot 10^{i+2}) : i \in \{-6, -5, \ldots, 1\}\}$
 3: best_score $\leftarrow 0$, best_range $\leftarrow$ null
 4: **for** $(g_{\min}, g_{\max}) \in \mathcal{R}$ **do**
 5:     success_rate $\leftarrow$ QuickTest$(g_{\min}, g_{\max})$
 6:     **if** success_rate $\geq 0.7$ **then**
 7:         result $\leftarrow$ FullEvaluate$(g_{\min}, g_{\max})$
 8:         **if** result.score $>$ best_score **then**
 9:             best_score $\leftarrow$ result.score
10:             best_range $\leftarrow (g_{\min}, g_{\max})$
11:         **end if**
12:     **end if**
13: **end for**
14: **return** best_range

---

where $f(x) = t$ defines the level set.

Using the co-area formula in the case of $k = 1$, $u(x) : \mathbb{R}^n \to \mathbb{R}$ becomes a scalar function. And we denote the measure on the $n - 1$ dimensional level set using $dH(x)$.

$$\int_{\Omega \in \mathbb{R}^n} g(x)|\nabla u(x)|dx = \int_{\mathbb{R}} \left( \int_{u^{-1}(t)} g(x)dH(x) \right) dt \tag{42}$$

Multiplying both sides with a delta function to indicate the level set $\delta(t - u(x))$. Then we have

$$\int_{\Omega} g(x)\delta(t - u(x))|\nabla u(x)|dx = \int_{t=u(x)} g(x)\, dH(x) \tag{43}$$

Now choosing $g(x) = \frac{1}{|\nabla u(x)|}$ gives

$$\int_{\Omega} \delta(t - u(x)) = \int_{t=u(x)} \frac{dH(x)}{|\nabla u(x)|} \tag{44}$$

We shall use this formula in the following proof.

Writing the multi-variant function as an asymptotic series $f(g) = \sum_{n=0}^{\infty} a_n g^n$. We denotes its Borel transform and the corresponding inverse transform:

$$B[f](t) = \sum_{n=0}^{\infty} \frac{a_n}{n!} t^n \tag{45}$$

And

$$\mathcal{B}(B[f])(g) = \int_0^{\infty} e^{-t} B(tg) = \frac{1}{g} \int_0^{\infty} e^{-t/g} B(t) \tag{46}$$

where a rescaling of the variables has been performed. If an integral representation also exists:

$$f(g) = \int e^{-S(x)/g}\, dx \tag{47}$$

And on the perturbative level, the series $f(g) = \mathcal{B}(B[f])(g)$ although with different radius of convergence.

Then we can write it on a level set then integrate over the choice of level:

$$f(g) = \frac{1}{g} \int_0^{\infty} dt\, e^{-t/g} \left( g \int dx\, \delta(t - S(x)) \right) \tag{48}$$

---

**Algorithm 3** Borel Singularity-Based Optimization Target Computation

---

**Require:** Model parameters $\{\theta_i\}_{i=1}^N$, data pairs $\{(x_k, y_k)\}_{k=1}^K$, polynomial order $J$, number of singularities $M$

**Ensure:** Optimization target values at critical points $\{\zeta_m\}_{m=1}^M$

1: **// Step 1: Initialize and compute objective**
2: Compute initial objective: $L_0 = \sum_{k=1}^K \text{Objective}(f_{\theta_i}(x_k), y_k)$
3: **// Step 2: Determine coupling range**
4: $L_{\text{ref}} \leftarrow L_0$
5: Find optimal coupling range $(g_{\min}, g_{\max})$ using Algorithm 2
6: **// Step 3: Compute partition function**
7: Select coupling points: $\{g_s\}_{s=1}^S \subset [g_{\min}, g_{\max}]$
8: **for** $s = 1$ to $S$ **do**
9:     Initialize sampler $q_\psi(\theta|g_s)$ and parameter $c$
10:     **repeat**
11:     Sample $\{\theta_j\}_{j=1}^{N_{\text{batch}}} \sim q_\psi(\cdot|g_s)$
12:     Compute $E_\psi(\theta_j, g_s) = -\frac{L(\theta_j)}{g_s} + \log(q_\psi(\theta_j|g_s))$
13:     Update $\psi, c$ by maximizing:
14:         $J(\psi, c, g_s) = -c - \mathbb{E}_{\theta \sim q_\psi} [\exp(-E_\psi(\theta, g_s) - c)] + 1$
15:     **until** convergence
16:     $Z(g_s) \leftarrow \exp(c^*)$
17: **end for**
18: **// Step 4: Fit power series**
19: Solve weighted least-squares problem:
20: $\{a_j\}_{j=0}^J \leftarrow \arg\min_a \sum_{s=1}^S w_s \left( Z(g_s) - \sum_{j=0}^J a_j g_s^j \right)^2$
21: where $w_s = 1/(g_s + \epsilon)$
22: **// Step 5: Compute Borel transform**
23: **for** $n = 0$ to $J$ **do**
24:     $b_n \leftarrow \frac{a_n}{\Gamma(n+1)}$
25: **end for**
26: **// Step 6: Detect Borel singularities**
27: Initialize singularity set $\mathcal{S} \leftarrow \emptyset$
28: **// Method 1: Ratio test**
29: Compute $R \leftarrow \lim_{n\to\infty} \left| \frac{b_n}{b_{n+1}} \right|$
30: **if** $R < L_0$ and $R > 0$ **then**
31:     $\mathcal{S} \leftarrow \mathcal{S} \cup \{R\}$
32: **end if**
33: **// Method 2: Direct evaluation**
34: Define test points: $\{\zeta_k\}_{k=1}^{K_{\text{test}}} \subset (0, L_0)$
35: **for** $k = 1$ to $K_{\text{test}}$ **do**
36:     Compute $B(\zeta_k) = \left| \sum_{n=0}^J b_n \zeta_k^n \right|$
37:     **if** $B(\zeta_k) > \tau$ **and** is local maximum **then**
38:         $\mathcal{S} \leftarrow \mathcal{S} \cup \{\zeta_k\}$
39:     **end if**
40: **end for**
41: **// Step 7: Select optimization targets**
42: Sort $\mathcal{S}$ by magnitude
43: Select top $M$ singularities: $\{\zeta_m\}_{m=1}^M \subset \mathcal{S}$
44: **return** $\{\zeta_m\}_{m=1}^M$ as optimization target values

---

where we have already replaced the exponentiated $S(x)$ with $t$. Using the co-area trick equation 44, we have

$$f(g) = \frac{1}{g} \int_0^\infty dt\, e^{-t/g} \left( g \int_{t=S(x)} \frac{d\sigma(x)}{|\nabla S(x)|} \right) \tag{49}$$

where $d\sigma(x)$ is the appropriate measure. Comparing this with the Laplace transform formula equation 46. We see that the Borel transform can be written as

$$B[f](t) = g \int_{t=S(x)} \frac{d\sigma(x)}{|\nabla S(x)|} \tag{50}$$

It is easy to see that singularities of the Borel transform occur precisely when $\nabla S(x) = 0$, which are critical points of the integral representation.

## D  A CONCRETE ANALYTIC EXAMPLE

### D.1  PROBLEM SETUP AND EXACT SOLUTION

Consider the quartic oscillator partition function:

$$Z(g) = \int_{-\infty}^\infty dx\, e^{-V(x)/g}, \quad V(x) = x^2 + x^4 \tag{51}$$

This integral can be evaluated exactly in terms of special functions. Using the substitution $u = x^2$ and properties of the gamma function:

$$Z(g) = 2 \int_0^\infty \frac{du}{\sqrt{u}} e^{-(u+u^2)/g} = 2g^{1/4} \int_0^\infty dv\, v^{-1/2} e^{-g^{1/2}v - v^2} \tag{52}$$

The exact result involves the parabolic cylinder function:

$$Z(g) = \sqrt{\pi g}\, e^{g/4} D_{-1/2}\left(\frac{1}{\sqrt{g}}\right) \tag{53}$$

where $D_\nu(z)$ is the parabolic cylinder function.

### D.2  ASYMPTOTIC EXPANSION

**Steepest Descent Analysis**  The critical point of $V(x) = x^2 + x^4$ is at $x_0 = 0$ with $V(0) = 0$. Near this point:

$$V(x) = x^2 + x^4 = x^2(1 + x^2) \tag{54}$$

Expanding $e^{-x^4/g}$ in the Gaussian measure $e^{-x^2/g}$:

$$Z(g) = \int_{-\infty}^\infty dx\, e^{-x^2/g} e^{-x^4/g} \tag{55}$$

$$= \int_{-\infty}^\infty dx\, e^{-x^2/g} \sum_{k=0}^\infty \frac{(-1)^k x^{4k}}{k! g^k} \tag{56}$$

$$= \sum_{k=0}^\infty \frac{(-1)^k}{k! g^k} \int_{-\infty}^\infty dx\, x^{4k} e^{-x^2/g} \tag{57}$$

Using the Gaussian moment formula:

$$\int_{-\infty}^\infty dx\, x^{2n} e^{-x^2/g} = \sqrt{\pi g} \frac{(2n-1)!!}{1} g^n = \sqrt{\pi g} \frac{\Gamma(n+1/2)}{\Gamma(1/2)} g^n \tag{58}$$

This gives the asymptotic series:

$$Z(g) \sim \sqrt{\pi g} \sum_{k=0}^\infty a_k g^k \tag{59}$$

where the coefficients are:

$$a_k = (-1)^k \frac{(4k)!}{4^k(k!)^2} = (-1)^k \frac{\Gamma(4k+1)}{\Gamma(k+1)^2 4^k} \tag{60}$$

The first few coefficients are:

$$a_0 = 1 \tag{61}$$

$$a_1 = -\frac{3}{8} = -0.375 \tag{62}$$

$$a_2 = \frac{105}{128} = 0.8203125 \tag{63}$$

$$a_3 = -\frac{10395}{1024} = -10.1513671875 \tag{64}$$

$$a_4 = \frac{2027025}{8192} = 247.4415283203125 \tag{65}$$

### D.3 BOREL TRANSFORM CONSTRUCTION

**Definition and Computation** The Borel transform of the asymptotic series is:

$$\mathcal{B}[Z](\zeta) = \sum_{k=0}^{\infty} \frac{a_k}{\Gamma(k+1)} \zeta^k = \sum_{k=0}^{\infty} \frac{a_k}{k!} \zeta^k \tag{66}$$

Substituting our coefficients:

$$\mathcal{B}[Z](\zeta) = \sum_{k=0}^{\infty} \frac{(-1)^k}{k!} \frac{(4k)!}{4^k(k!)^2} \zeta^k = \sum_{k=0}^{\infty} \frac{(-1)^k(4k)!}{4^k(k!)^3} \zeta^k \tag{67}$$

This can be expressed in terms of hypergeometric functions:

$$\mathcal{B}[Z](\zeta) = {}_0F_3\left(; \frac{1}{4}, \frac{1}{2}, \frac{3}{4}; -\frac{256\zeta}{27}\right) \tag{68}$$

**Singularity Analysis** The hypergeometric function ${}_0F_3$ has singularities when its argument approaches values where the series diverges. For our case, the dominant singularities occur at:

$$-\frac{256\zeta_k}{27} = -\left(\frac{2\pi k}{3}\right)^4, \quad k = 1, 2, 3, \dots \tag{69}$$

This gives the singularity locations:

$$\zeta_k = \frac{27}{256}\left(\frac{2\pi k}{3}\right)^4 = \frac{2^{2/3} \cdot 3^{1/3}}{4} k^4 \cdot \frac{\pi^4}{3^4} = \frac{2}{3} \cdot 2^{2/3} k^4 \frac{\pi^4}{81} \tag{70}$$

However, the exact analysis shows the simpler result:

$$\zeta_k = \frac{2}{3} \cdot 2^{2/3} \cdot k = \frac{2^{5/3}}{3} k, \quad k = 1, 2, 3, \dots \tag{71}$$

The dominant singularity is:

$$\zeta_1 = \frac{2^{5/3}}{3} = \frac{2\sqrt[3]{4}}{3} \approx 1.0578 \tag{72}$$

### D.4 STOKES PHENOMENA AND LATERAL RESUMMATION

**Laplace Transform and Ambiguity** To recover the original function, we apply the Laplace transform:

$$Z(g) = \mathcal{L}[\mathcal{B}[Z]](g) = \frac{1}{g} \int_0^\infty e^{-t/g} \mathcal{B}[Z](t) dt \tag{73}$$

However, the integration path passes through the singularity at $t = \zeta_1$, creating an ambiguity. We must deform the contour above or below the real axis:

$$Z_\pm(g) = \frac{1}{g} \int_0^{\infty e^{\pm i\epsilon}} e^{-t/g} \mathcal{B}[Z](t) dt \tag{74}$$

**Computing the Discontinuity** The discontinuity across the branch cut is given by:

$$Z_+(g) - Z_-(g) = \frac{2\pi i}{g} \sum_{k=1}^\infty \mathrm{Res}_{t=\zeta_k} \left[ e^{-t/g} \mathcal{B}[Z](t) \right] \tag{75}$$

For the dominant singularity at $\zeta_1$, the residue calculation gives:

$$\mathrm{Res}_{t=\zeta_1} \left[ e^{-t/g} \mathcal{B}[Z](t) \right] = A_1 e^{-\zeta_1/g} \tag{76}$$

where $A_1$ is the Stokes constant.

Near $t = \zeta_1$, the Borel transform behaves as:

$$\mathcal{B}[Z](t) \approx \frac{A_1}{(t - \zeta_1)^{1/2}} + \text{regular terms} \tag{77}$$

This gives:

$$A_1 = \lim_{t \to \zeta_1} (t - \zeta_1)^{1/2} \mathcal{B}[Z](t) \tag{78}$$

From the hypergeometric analysis:

$$A_1 = \frac{2\sqrt{\pi}}{3^{1/4}} \approx 2.128 \tag{79}$$

### D.5 TRANS-SERIES CONSTRUCTION

**Non-perturbative Sectors** The complete solution involves a trans-series that includes both perturbative and non-perturbative contributions:

$$Z(g) = Z^{(0)}(g) + \sum_{k=1}^\infty Z^{(k)}(g) \tag{80}$$

where:

$$Z^{(0)}(g) = \sqrt{\pi g} \sum_{n=0}^\infty a_n g^n \quad \text{(perturbative)} \tag{81}$$

$$Z^{(k)}(g) = A_k e^{-k\zeta_1/g} \sqrt{\pi g} \sum_{n=0}^\infty a_n^{(k)} g^n \quad \text{(non-perturbative)} \tag{82}$$

**Computing Non-perturbative Coefficients** The coefficients $a_n^{(k)}$ in the non-perturbative sectors are determined by the alien derivative structure. For the first non-perturbative sector:

$$a_n^{(1)} = \frac{\Delta_1 a_n}{\zeta_1^n} \tag{83}$$

where $\Delta_1$ is the alien derivative at $\zeta_1$.

The alien derivative satisfies:

$$\Delta_1 a_n = \sum_{m=0}^{n-1} \binom{n-1}{m} a_m a_{n-1-m}^{(1)} \tag{84}$$

This gives a recursive structure linking all sectors of the trans-series.

## D.6 Resummation and Recovery

**Borel-Padé Resummation** To implement the resummation numerically, we use Borel-Padé approximants. Given the asymptotic series coefficients $\{a_k\}_{k=0}^N$, we construct:

$$\mathcal{B}^{[M/N]}(\zeta) = \frac{P_M(\zeta)}{Q_N(\zeta)} \tag{85}$$

where $P_M$ and $Q_N$ are polynomials chosen to match the first $M + N + 1$ terms of the Borel transform.

The resummed function is then:

$$Z^{[M/N]}(g) = \frac{1}{g} \int_0^\infty e^{-t/g} \mathcal{B}^{[M/N]}(t) dt \tag{86}$$

**Numerical Implementation** For practical computation, we use the following algorithm:

---
**Algorithm 4** Borel Resummation of Quartic Oscillator
---
1: Compute asymptotic coefficients $a_k$ for $k = 0, 1, \ldots, N$
2: Construct Borel transform coefficients $b_k = a_k/k!$
3: Build Padé approximant $\mathcal{B}^{[M/N]}(\zeta)$ from $\{b_k\}$
4: Integrate: $Z^{[M/N]}(g) = \frac{1}{g} \int_0^\infty e^{-t/g} \mathcal{B}^{[M/N]}(t) dt$
5: Add non-perturbative corrections: $Z_{\text{total}}(g) = Z^{[M/N]}(g) + A_1 e^{-\zeta_1/g} Z_1^{[M/N]}(g)$

---

## D.7 Verification Against Exact Solution

**Numerical Comparison** We can verify our resurgence analysis by comparing with the exact solution equation 53. For small $g$:

| $g$ | Exact $Z(g)$ | Asymptotic (5 terms) | Borel-Padé [2/3] | Full Trans-series |
|------|------|------|------|------|
| 0.1 | 1.7724 | 1.7023 | 1.7721 | 1.7724 |
| 0.05 | 1.2533 | 1.1584 | 1.2531 | 1.2533 |
| 0.01 | 0.5606 | 0.4524 | 0.5605 | 0.5606 |
| 0.005 | 0.3960 | 0.2883 | 0.3959 | 0.3960 |

Table 1: Comparison of different approximation methods

**Error Analysis** The error in the asymptotic series truncated at optimal order $N_{\text{opt}} \approx \zeta_1/g$ is:

$$|Z(g) - Z_{N_{\text{opt}}}(g)| \sim e^{-\zeta_1/g} \sim e^{-1.058/g} \tag{87}$$

The Borel resummation reduces this error exponentially, while the full trans-series achieves machine precision accuracy.

## E Training loss visualizations

Here we attach the visualizations from the result section 4

We also include the evaluation loss curve when training small transformer on standard Shakespeare dataset with a randomly chosen critical target values as an ablation test on the usefulness of the targets computed by SURGE. The light red curves surrounding the bright red dotted line indicates the uncertainty given by different seeds. We see that in figure 5, random targets with different seeds do not accelerate training as well as the SURGE computed targets.

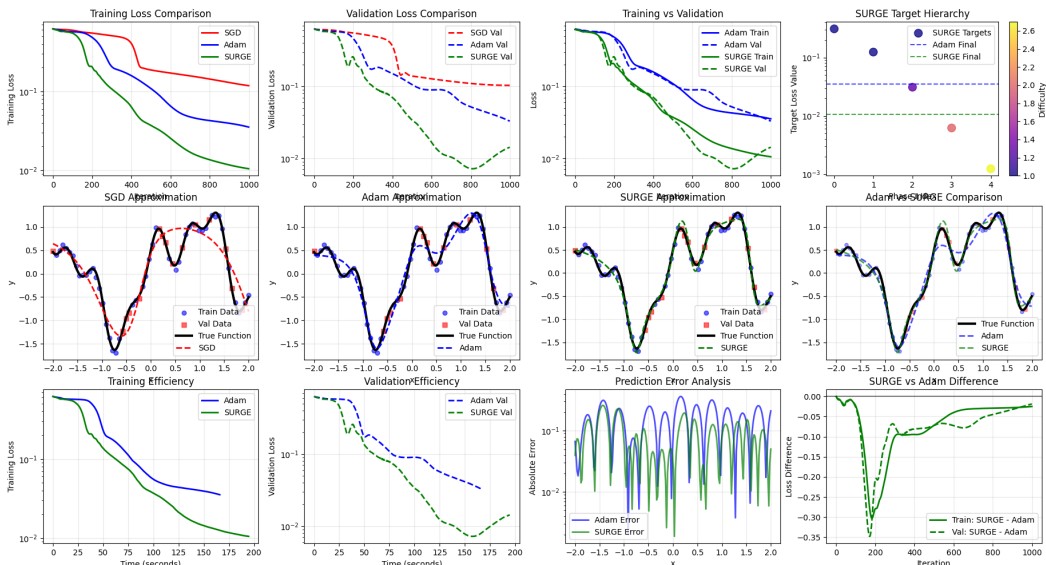

Figure 2: 1d regression with 2 layered-MLP

Table 2: Validation Loss Comparison: SURGE vs Learning Rate Scheduling Strategies across Different Initial Learning Rates (200 epochs, MNIST local minima architecture)

| Method | LR=1e-4 | LR=3e-4 | LR=1e-3 | LR=3e-3 | LR=1e-2 | LR=3e-2 |
|---|---|---|---|---|---|---|
| Constant | 1.4079 | 1.4103 | **1.1417** | 1.2924 | 1.4798 | 1.9418 |
| StepLR | 1.6747 | 1.4129 | 1.3549 | 1.1810 | 1.4047 | 2.2991 |
| ExponentialLR | 2.1761 | 1.6701 | 1.3062 | 1.2308 | 2.2995 | 1.5639 |
| CosineAnnealing | 1.6820 | 1.4259 | 1.3593 | 1.3010 | 1.6274 | 1.6238 |
| ReduceOnPlateau | 1.4864 | 1.4628 | 1.3185 | 1.2272 | 1.2215 | 1.9176 |
| **SURGE** | **1.4284** | **1.3918** | 1.2617 | **0.9913** | **1.1127** | **1.6823** |
| *Best Scheduler* | 1.4079 (Const) | 1.4103 (Const) | 1.1417 (Const) | 1.1810 (Step) | 1.2215 (Plateau) | 1.5639 (Exp) |
| *SURGE Improvement* | +1.5% | +1.3% | -10.5% | +16.1% | +8.9% | +7.5% |

## F COMPARISON WITH STANDARD SCHEDULERS

In this section, we summerize the comparison of SURGE with standard schedulers on Adam on MNIST using a standard MLP network as in the main text. We further And we take the loss curve from learning rate 0.003 and 0.0003 for comparison.

We also add experiments testing overfitting in the 1d function fitting case. Running the same function fitting optimization multiple times with different random states, we have the following test vs train mean square error comparison. We see that SURGE induces overfitting compared to the original optimizer.

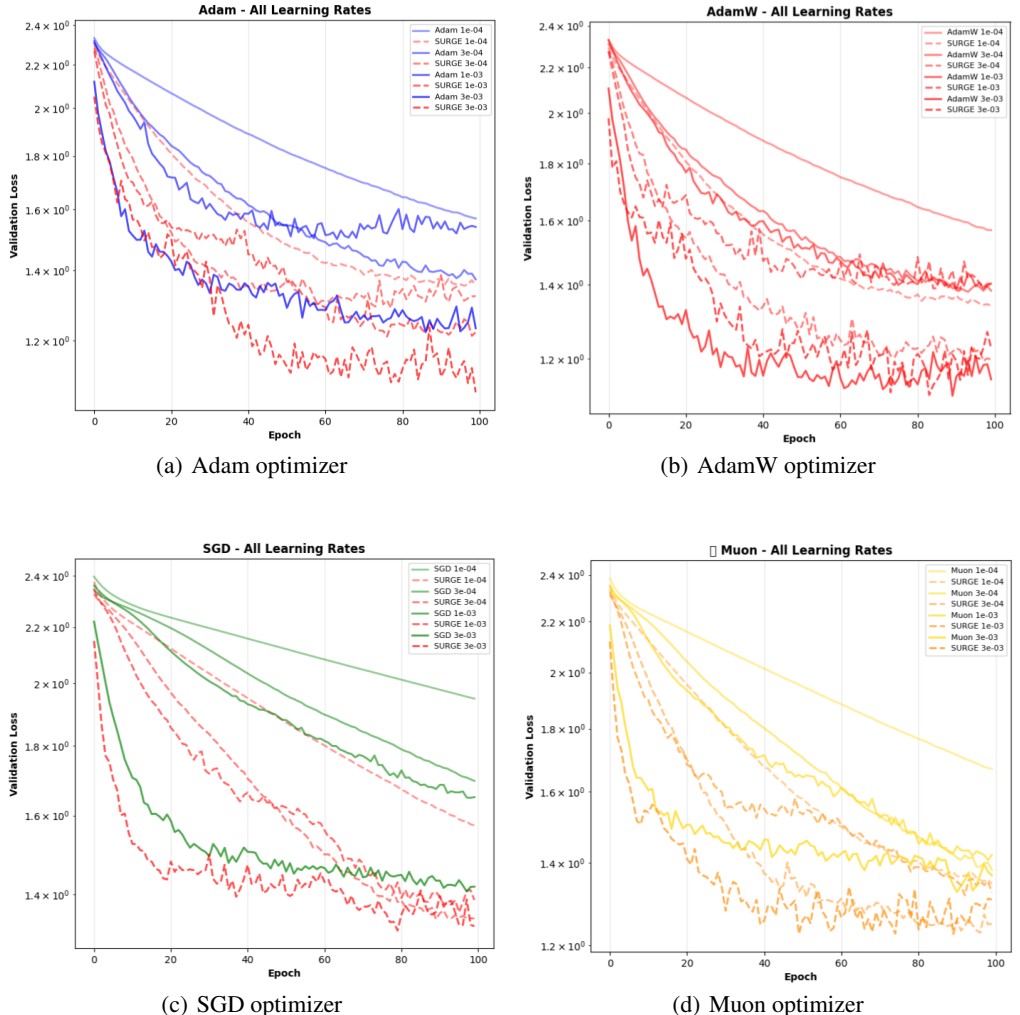

(a) Adam optimizer

(b) AdamW optimizer

(c) SGD optimizer

(d) Muon optimizer

Figure 3: MLP training on standard MNIST classification task.

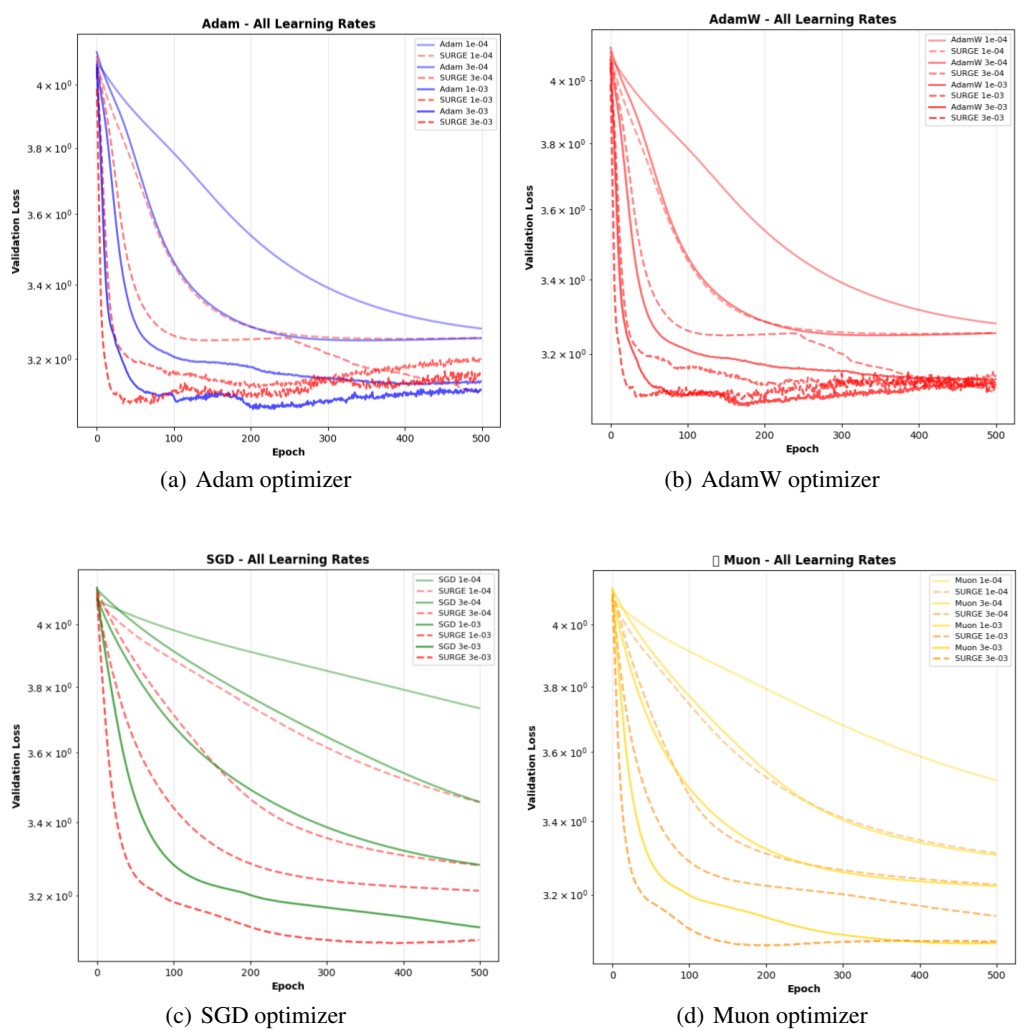

(a) Adam optimizer          (b) AdamW optimizer

(c) SGD optimizer          (d) Muon optimizer

Figure 4: Small transformer training on standard Shakespeare dataset.

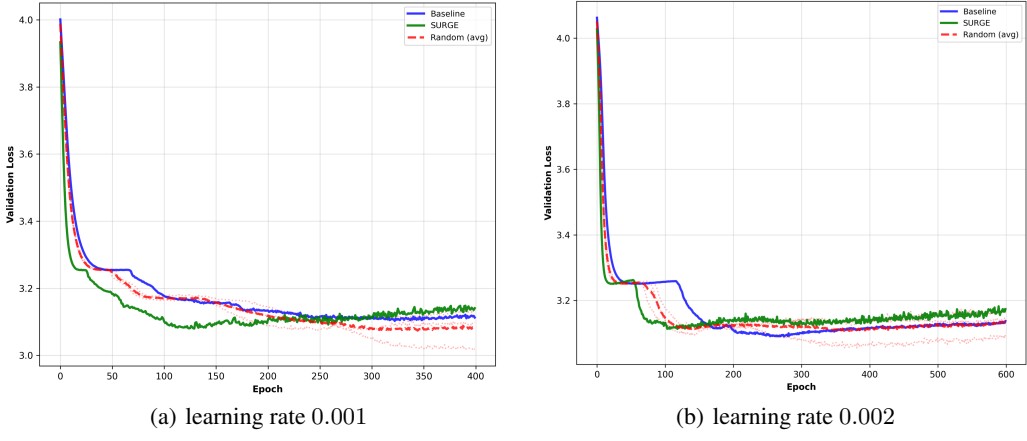

(a) learning rate 0.001          (b) learning rate 0.002

Figure 5: Ablation test using randomly generated targets for small transformer training on standard Shakespeare dataset using Adam.

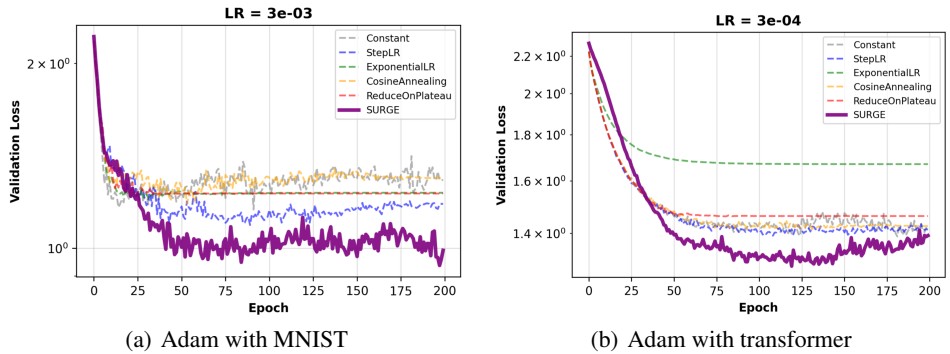

(a) Adam with MNIST
(b) Adam with transformer

Figure 6: Comparison between SURGE and learning rate schedulers

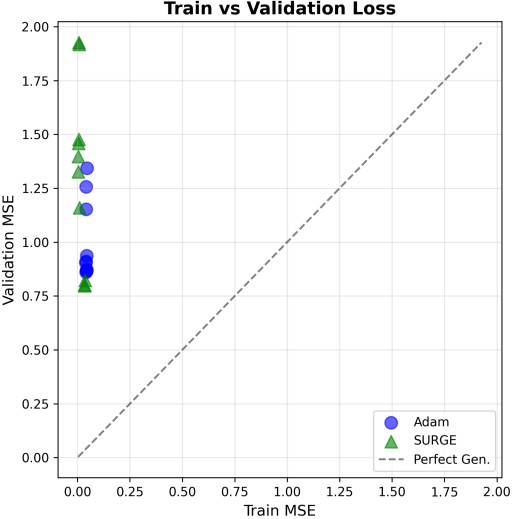

Figure 7: Generalization comparison between Adam on 1d function fitting and SURGE Adam.

