# OpenReview forum: "Globally aware optimization with resurgence"
_ICLR.cc/2026/Conference — ICLR 2026 Poster_

### Official Review · Reviewer_fviL · 2025-10-28

**Soundness:** 2
**Presentation:** 1
**Contribution:** 2
**Rating:** 2
**Confidence:** 4

**Summary:**

This work proposes to inject non-local informtaion about the objective function into the optimization process.
It relies on resurgenece theory to obtain the function values of the critical points of the objective function and then uses this information for learning rate adaptation.

**Strengths:**

As far as I can tell, the use of resurgence theory to improve optimization algorithms is novel and this idea seems to have a lot of potential. If one can indeed obtain a god approximation function values of the critical points of the objective function, this could dramatically improve optimization algorithms.


This work provides a thorough introduction on this topic.

**Weaknesses:**

### Lack of connection between "2.1 An intuitive intro to resurgence" and the rest of the paper
This work starts with a lengthy introducion to resurgence theory. However, it is not clear how large parts of this introduction relate to the rest of the paper. As far as I understand, the discussion about the integration contour 172-205 is not necessary to understand the rest of the paper and only confuses the reader.

### Unclear what is new vs existing and missing references
- There should be a reference for Thm. 1
- I don't understand whether Prop. 2 and Thm. 3 are novel or existing results. If they are existing, please provide references. Also, both the statement and the proof of Thm. 3 need much more details: how is the integration over the level set done for instance?

### General lack of literature review
Optimization on non-convex landscapes as well as adaptive learning rate schemes are both very active research areas but the discussion of related work is very limited.
In particular, the following topics should be discussed:
- Adaptive and parameter-free leanring rates, with a particular focus Polyak-type schemes, which also use function values to adapt the learning rate.
- Langevin dynamics on non-convex landscapes: these methods sample from the Gibbs distribution, which is closely related to the statistical mechanics discussion in Sec. 2.3.
- Simulated annealing and tempering methods, which also rely on the partition function to explore non-convex landscapes and therefore seem related to the proposed approach.

### Unclear how the practical algorithm follows from the theory
Most of the steps of the algorithm in 3.1 appear pretty ad-hoc and it is not clear why the authors chose these particular steps.
- In particular, at l32, why choose the maximum value in T lower than the current function value? Why not the minimum? Indeed, if the theory is correct, the set T contains the function values of all critical points: therefore one should only choose the minimum one, since it corresponds to the global minimum of the objective function. This discrepancy between theory and practice is never discussed.
- Why use this function value to adapt the learning rate as in l34? Why not use it in other ways? In particular, why not use a Polyak-type learning rate scheme?

### Experimental section is weak
All experiments are on toy problems (1D function or MNIST), hyperparameters are not proeprly tuned (one should report the result for the best learning rate for each method), there are no error bars, no adaptive learning rate baselines (e.g., Polyak-type schemes at the very least) and no clear benefit of the proposed method is shown.

Moreover, it is not clear whether the resurgence theory is indeed useful for the proposed algorithm: one could imagine much simpler approaches.
For instance at L342, this work explains how to obtain an approximation of the partition function
$$
Z(g) = \int e^{-f(\theta)/g} d\theta
$$
which I agree is an interesting idea. However, the authors then use the intricate machinery to exract the function values of the critical points from this partition function while, if I am not mistaken, $- g \log Z(g)$ for small $g$ is already a good approximation of the global minimum of $f$. This value could then be used in a Polyak-type learning rate scheme or in the proposed learning rate adaptation scheme. It would be interesting to see whether this simpler approach, that does not use resurgence theory, works just as well as the proposed method.

### Not clear how the proposed scheme helps escaping "suboptimal regions"
The abstract states that "These target values provide global guidance to local optimizers, enabling [...] escape from suboptimal regions". However, it is not clear how the proposed learning rate adaptation scheme helps escaping suboptimal regions. If the function value of a critical point is used to adapt the learning rate, and if the current iterate is in a basin of attraction of a suboptimal local minimum, then the adapted learning rate will likely be small (since the function value at the current iterate is close to that of the suboptimal local minimum). Therefore, it is not clear how this scheme helps escaping suboptimal regions.

**Questions:**

Please address "Lack of connection between "2.1 An intuitive intro to resurgence" and the rest of the paper", "Unclear what is new vs existing and missing references" and "General lack of literature review" above.

---

> ### Author Response · Authors · 2025-11-12
>
> We first thank the reviewer for their time reading the manuscript and providing detailed feedbacks. Since the current existing methods of analyzing optimization landscape globally are quite scarce, our intention is to give a first attempt in introducing the resurgence tools from mathematical physics. Admittedly the current practically tractable usage of the resurgence framework in the paper is crude, we intend to further this endeavour and improve the engineering of the algorithms.
>
>
> 1. Section 2.1 was trying to make an effort to start off using simple examples to introduce the concept and philosophy of resurgence, practically introducing theorem 1 with examples. Especially 172-205 which is later applied to the partition function of an ensemble of neural networks in showing that its factorially divergent power series expansion contains useful information about the critical values of the loss function later in appendix C. The contour integral part is the key step which allows one to convert a high dimensional parameter space computation to a complex analysis problem via Borel transform. The contour integral picks up singularities that correspond precisely to the critical values of the loss.
>     We shall include some further explanations to the contour integral usage to connect the flow better.
>
> 2. We thank the reviewer for picking up on the reference issues. We acknowledge and add the missing references. Theorem one is the main result by Escalle, which was cited a few times before, we add additional citation next to theorem 1. Proposition 2 and Theorem 3 are partially existing and partially new, we have only cited the exiting results on L55. We shall add citation directly next to them to make it clear.
>
> 3. We appreciate that there is a gap in the literature review, we add further paragraphs for a more comprehensive review of existing literature on the three topics listed by the reviewer.
>
> 4. The algorithm is chosen mainly for practical implementation purposes. A maximum is chosen for the following reason. In high dimensional optimziation landscape, the number of saddle point is very large, practically it is impossible to compute all of them, we enforce a gap in the list of targets we compute at the beginning. Secondly, empirically, if we start with the smallest critical value to start with, the scaling on the learning rate is quite large, this makes the training process unstable. For example even in figure 3c, SURGE worsens the performance of SGD with learning rate 1e-3 because of this instability.
>     The learning rate scaling is admittedly a rather crude use of the targets computed using resurgence. Practically it embodies the simple idea that if you are far from the next critical loss value, pick up the pace, and vice versa. We have not incorporated any use of learning rate scheduling or scheme for both the base optimizer and the SURGE wrapper. The effect of those are worth exploring further.
>
> 5. Regarding the experiments, the main part was the MLP on MNIST and transformer on Shakespeare dataset, which was compared across a spectrum of learning rates for all the optimizers considered. They are still rather toy examples admittedly, they serve more as a proof of concept here. The partition function $Z(g)$ is not a function of the parameters $\theta$ itself, hence $-g\log Z(g)$ is certainly not $f$, the integral over $\theta$ is crucial. The reason the partition function was introduced is to use the resurgence trick to obtain global information on the critical value of the function $L$. Hope this clarifies things further.
>
> 6. "and if the current iterate is in a basin of attraction of a suboptimal local minimum, then the adapted learning rate will likely be small (since the function value at the current iterate is close to that of the suboptimal local minimum)", regarding this comment, what we do in the algorithm is to change targets when the current loss is within a certain threshold distance from the current target loss value. This ensures that there's always some distance between the target and the current loss value. Another point is that we never downscale the learning rate, when the current loss is close to the target, we fall back onto the original optimizer.

---

> > ### Comment · Reviewer_fviL · 2025-11-23
> >
> > I thank the authors for their detailed answer.
> >
> > * **On the contour integral 175–205**: I still do not understand why this part is necessary. All the tools required for the algorithm appear to already be in Section 2.2, and are introduced in Section 2.1 up to line 174. As written, the contour argument feels disconnected from the rest of the paper.
> >
> > * **On Polyak step-size**: Thank you for adding relevant references. However, I disagree with the characterization “adapt to local curvature information” and “remain blind to global geometry.” The point of these methods is precisely to use the function value at (or an estimate of) the global optimum to guide optimization. This is conceptually very close to SURGE, which uses the function values of all critical points. Given this strong conceptual overlap, these methods should be properly discussed and compared against.
> >
> > * **On escaping suboptimal regions**: In your response, you mentioned several crucial algorithmic mechanisms (threshold for switching targets, fallback to the base optimizer) that are not clearly described in the paper. These should be made explicit.
> >
> > * **On the experiments**: Even for proof-of-concept purposes, experiments should be conducted properly—tuned hyperparameters for each method, reasonable baselines, and multiple random seeds. The central issue for me is that the current results do not demonstrate that the full resurgence machinery provides practical benefit: (i) performance is not clearly better than the baselines shown, (ii) only a few baselines are included, and (iii) it is unclear whether improvements, when present, come from resurgence or simply from the update rule in (34). This last question should be answered with ablations: e.g., what if one uses only the target $-g \log Z(g)$? Or a simple grid between $0$ and $f(x_0)$? Without such comparisons, the empirical value of resurgence remains uncertain.

---

> ### Author Response · Authors · 2025-11-23
>
> Thank you to the reviewer for further questions, we appreciate the effort to help shaping the paper better. Hopefully my following comments clear things up further.
>
> 1. The contour integral was the entire point of the mathematical theory of resurgence. Up to 174, simply switching integral and a divergent sum results in the integrand blowing up at a point on the integration domain. This is because the switching move was illegal, to resolve this, one is forced to complexify and do contour integral to recover the original series via Laplace transform.
> 2. The Polyak step-size algorithm, indeed adjusts the learning rate according to optimum value of the objective function. In the case when the optimum values of the objective function are known (such as when objective function is convex or simple enough), I agree with the comment and shall revise the statement in the paper. The point of SURGE however, is to approximate these optimum values of the objective function even when it’s complicated and non-convex. So they carry fundamentally different purpose to begin with.
> 3. We thank the reviewer for this request, we shall add further comments in the paper.
> 4. Hyperparameter tuning was done on the small models given the compute we have access to. Performance on various different learning rates have been included already, we intend to further experimental side to include more datasets and architectures and even more optimization problems given more compute resources and time. Regarding the last question, we can add a further ablation test that uses grid targets and compare, this will be included shortly.
>
> We also just wanted to further clarify this again, the purpose of this paper is to introduce the tool of resurgence and implement as an algorithm of finding global critical values of objective functions (for generic objective functions). The usage as a wrapper on optimisers during neural net training is one way of using the set of objective function critical values.

---

> > ### Comment · Reviewer_fviL · 2025-11-25
> >
> > I thank the authors for their detailed reply; here are my answers to their comments:
> > - **For the contour integral:** I agree that, under the hood, the contour integral is necessary to make the mathematics of resurgence work. However, this is not needed to understand the method. Nonetheless, this was just a minor comment on my end; thank you for clarifying.
> > -  **On the experiments:** I am not criticizing the scale of the experiments, only the execution and reporting. For instance, instead of reporting the performance for every step-size, it would be better to select a step-size on a validation set and report test/train losses, error bars, etc.
> > - **Finally, on the purpose of the paper:** thank you for this clarification. You mention that "the purpose of this paper is to introduce the tool of resurgence and implement it as an algorithm of finding global critical values of objective functions". However, this is not at all reflected in the paper: for instance, both the title “Globally aware optimization with resurgence” and the fact that the paper is focused on “SURGE: Singularity Unified Resurgent Gradient Enhancement”, which is an optimization method (see Algorithm 1), strongly suggest that this paper is about improving optimization algorithms. I wholeheartedly agree that the idea of using resurgence to obtain global information is interesting, but more work is needed before publication. At present, the contributions are mainly limited to introducing the method for extracting critical values (especially since the authors acknowledged that most theoretical results are existing), without convincing empirical verification or a clear application demonstrating its practical usefulness.

---

> > > ### Author Response · Authors · 2025-11-25
> > >
> > > Thank you for further clarifications.
> > >
> > > The current loss curves across different learning rates shown are already validation loss on 20% of the respective datasets. We are more than happy to include further detail in appendix F to include those details as a table for some specific experiments.
> > >
> > > The existing literature on resurgence only concerns the quantum field theory perturbative methods, used to resum contributions to scattering probabilities of particles. The main theorem 3 is partially mentioned (the measure theory co-area formula) in the cited paper, theoretically we complete that proof and use it on ensemble of parameter optimization. Practically, we come up with the algorithm of estimating global critical values of the objective function. We understand the reviewer’s angle since the title of the paper made it seem more like an optimization centric paper. We are happy to change the pivot of the narrative to the novelty of the algorithm using mathematical physics techniques and only using the estimated critical values for optimization guidance. (For example the title as “Resurgence for estimation of global critical values of objective function”) This was our original purpose and apologies if the writing made. It less clear.

---

### Official Review · Reviewer_Zpez · 2025-10-29

**Soundness:** 3
**Presentation:** 3
**Contribution:** 4
**Rating:** 8
**Confidence:** 4

**Summary:**

The paper proposes SURGE (Singularity Unified Resurgent Gradient Enhancement), a novel optimization framework that leverages resurgence theory (via Borel transforms of partition functions over neural-network parameter space) to identify critical objective values in the loss landscape. These values are used to guide adaptive learning rates for any base optimizer, with the aim of improving convergence and escape from sub-optimal local minima. The authors provide theoretical arguments linking Borel singularities of the network partition function to landscape critical values, and demonstrate empirical improvements (e.g., Sec. 4.2, ~10K-parameter networks) compared to baseline optimizers.

The approach is conceptually intriguing and mathematically elegant, and has potential to open a new line of optimization research. However, I feel that the current version may fall short in practical guidance, sensitivity to numerical computations, scalability evidence, and empirical benchmarking.

**Strengths:**

(1) From my point of view, the use of resurgence theory and partition-function singularity detection to gain global landscape information is highly innovative in ML optimization work.
(2) The algorithmic wrapper (SURGE) is optimizer-agnostic and shows promising empirical results.
(3) The connection between complex-analysis tools (Borel transforms) and ML optimization adds theoretical depth and opens a new direction of research.

**Weaknesses:**

(1) 	The method hinges on computing a partition function and identifying singularities to yield target values; however, the manuscript lacks guidance on how to choose the initial model parameters or coupling parameter g in general settings (even simple losses like cross-entropy or MSE). The authors should provide practical heuristics or guidelines for parameter initialization.
(2) 	Computing the partition function Z(g) over a coupling range and performing Borel analysis seem to be likely very expensive and may restrict the method’s applicability. I feel that the method may simply shift the optimization burden onto a heavy pre-computation. More discussion is needed on the cost and scalability trade-offs. In addition, I wonder how the quality of numerical approximation of asymptotic series, equation (29), affect the overall performance of adapting learning step when employing (33) and (34).
(3)  Although the method is theoretically rich, the empirical setting (10K parameters) is still modest relative to real-world deep networks (millions of parameters).  Also, the authors note that SURGE may amplify over-fitting by pushing harder toward identified critical values, but no empirical analysis or mitigation strategy is provided.

**Questions:**

(1) In addition, I wonder how the quality of numerical approximation of asymptotic series, equation (29), affect the overall performance of adapting learning step when employing (33) and (34).

(2) The paper reports improvement in final objective values, but lacks wall-clock time comparisons against standard optimizers.

(3) I wonder sensitivity analysis and performance under noisy given data of the proposed algorithm as a wrapper for general optimizer with adaptive learning rate.

---

> ### Author Response · Authors · 2025-11-12
>
> We first thank the reviewer for their time reading the manuscript and providing detailed feedbacks. Since the current existing methods of analyzing optimization landscape globally are quite scarce, our intention is to give a first attempt in introducing the resurgence tools from mathematical physics. Admittedly the current practically tractable usage of the resurgence framework in the paper is crude, we intend to further this endeavour and improve the engineering of the algorithms.
>
>
> 1. We indeed not provided a guide for initialization of the network parameters, we have used standard initialization, although the framework works generically for any initialization, it would be nice to have a characterization of advantages of different initializations by running resurgence analysis on different initializations. We note this suggestion.
>
> 2. Indeed estimation of the partition function is the key bottleneck of the formalism, the complexity of the algorithm is $\mathcal{O}(N^2Bp)$ where $N$ is the order of derivatives computed, $p$ is the network parameter and $B$ is the batch size. So it is linear in network size, parallelization and other tricks are under exploration for scaling. The quality of numerical approximation $(29)$ indeed adds another layer of uncertainty to the estimation, however practically it is not that difficult to estimate the coefficients, since we know they grow factorially with the expansion order $n$, which just leaves some value $b$ in $a_n\sim b^n n! $ to be estimated.
>
> 3. Indeed we have not tested the actual test set performance of the optimizers compares with their SURGE counterparts at large scale, the statement comes from the 1d function fitting case where the SURGE fitted function is sometimes observed to have larger variance compared to standard optimizers. SURGE allows an optimization process to quickly approach a low training loss on the objective function but may induce overfitting. We made that intuitive comment not as a confirmed statement.
>
> 4. Could you please clarify this question further? "(3) I wonder sensitivity analysis and performance under noisy given data of the proposed algorithm as a wrapper for general optimizer with adaptive learning rate."

---

> > ### Comment · Reviewer_Zpez · 2025-11-12
> >
> > -- Regarding the question: (3) I wonder sensitivity analysis and performance under noisy given data of the proposed algorithm as a wrapper for general optimizer with adaptive learning rate.
> >
> > I would like to see the numerical performance comparison between
> > (a) the proposed algorithm (SURGE) as a wrapper for general optimizer with adaptive learning rate;
> > vs
> > (b) a general optimization scheme with adaptive learning rate
> >
> > when optimizing the same high-dimensional benchmark function such as the Schwefel's function, under same small random perturbation. I would like to see (1) if SURGE wrapper can still help find the global optimal and most local minimums vs perturbation; (2) running time verse dimension.
> >
> > -- Regarding 'the statement comes from the 1d function fitting case where the SURGE fitted function is sometimes observed to have larger variance compared to standard optimizers'
> >
> > I wonder if the authors can provide more quantitative results regarding the the amount of  variance increase when doing 1d or 2d function fitting. It would be great if the authors can provide a bound on the variance increase in terms of dimension.
> >
> > Thank you!

---

> > > ### Author Response · Authors · 2025-11-13
> > >
> > > Thanks for the reply! We've added an appendix showing two added experiments. 1. different optimizers with constant, stepLR, exponential LR, cosine annealing and reduce on plateau learning rate schedulers compared against with SURGE wrapper. 2. We provide generalization gap and variance of 1d function fitting case. 3. We are attempting to implement some experiments regarding this comment "when optimizing the same high-dimensional benchmark under same small random perturbation".

---

### Official Review · Reviewer_YZ6g · 2025-10-29

**Soundness:** 3
**Presentation:** 3
**Contribution:** 2
**Rating:** 4
**Confidence:** 3

**Summary:**

The authors propose a boost to local searching algorithms for objective functions that is able to aim critical points by exploiting global (although approximated) information about the function itself. They show how this improves time of convergence in a few examples.

**Strengths:**

It seems a truly original and intriguing piece of work, based on robust mathematical theory.
It is visible the effort towards clarity and very appreciated, however it remains mysterious the intuition about the algorithm proposed.

**Weaknesses:**

It is visible the effort towards clarity and very appreciated, however it remains mysterious the intuition about the algorithm proposed.
It is also missing a critical discussion on the limitations of this approach, despite the pretended global information obtained about the objective function. It cannot be the full information otherwise it would be overcoming NP hardness.
Without these discussions it is hard to understand what is the actual contribution to improve algorithms convergence in the general case, this explains my conservative rating FAIR to “contribution”.

**Questions:**

There must be limitations to this approach otherwise we should believe that it can make GD able to solve NP hard problems in polynomial time which it would be the astonishing main result, were it true, but it is not mentioned, therefore there must be something I do not understand. The authors should discuss how NP problems remain not solvable in polynomial time even with this approach and in general discuss limitations more explicitly, beside the advantages.

For rough objective functions the evaluation of the partition function should be exponentially hard. What is the cost expected also considering the approximation proposed?

SGD does not necessarily see the same landscape as the objective function.

Hitting critical points is not necessarily useful, especially when there is a huge number of saddles and maxima, and few minima which may be even fewer if associated to low values of the objective function. In fact it would be better if GD could avoid saddles, rather than hitting them, as they slow the dynamics down without giving other advantages. How can the algorithm select among the critical points?

It was appreciated the intuitive introduction to resurgence. It is way less clear the implementation of the algorithm. Although apparently explained in multiple places, I cannot find intuition on how it must work.

Line 170-171 “we see where it backfires at us in the example” which example? It should be made clearer.

In figure 1 on both sides there are 4 double series of data, while legends refers to only three choices of parameters. In some instances long time behaviour between blue and red is very similar, in other very different. What is the reason for that?

---

> ### Author Response · Authors · 2025-11-12
>
> We first thank the reviewer for their time reading the manuscript and providing detailed feedbacks. Since the current existing methods of analyzing optimization landscape globally are quite scarce, our intention is to give a first attempt in introducing the resurgence tools from mathematical physics. Admittedly the current practically tractable usage of the resurgence framework in the paper is crude, we intend to further this endeavour and improve the engineering of the algorithms.
>
> 1. The reviewer is indeed correct, and we address this around L397. The essence is that although one can obtain the critical value of the objective function, the set of parameters that attain that value is not known. The NP hard problem is the latter, we did not solve that.
>
> 2. Indeed it can be really difficult to estimate the partition function in high dimensions. We use the inequality to estimate it and train a separate neural network on that objective. Admittedly this adds an additional layer of complexity and quantifying it theoretically can be quite difficult as it depends on the precision one is after and the size/training time of this separate neural net.
>
> 3. Indeed SGD and other stochastic gradient methods only sees an averaged objective function within the batch, which may very well be a completely biased approximation of the true landscape. We do not have a good answer for this at the moment, besides claiming that practically, the approximation of the landscape within the batch seems good enough.
>
> 4. Currently the algorithm first computes a list of objectives values that are lower than the current loss (in high dimensions when number of saddles is large, we force a gap between the targets), then simply picks the largest among them, as the current loss is within a threshold of that target, it switches to the next largest on the list.
>
> 5. Indeed we do not have a theory for convergence guarantee of the algorithm, we aim to further characterize this.
>
> 6. Exchanging sum and integral is an illegal operation (results in something divergent equals something convergent) when the sum itself is divergent (which is the case here), we saw that following this, in L173-L176, the integrand of the real integral resulting from the switch gives us a pole on its integration domain, which is not computable, hence the complexification later.
>
> 7. Thanks for picking the typo up, we missed a legend marker in the plot. Empirically, what we noticed is that SURGE can be quite efficient in the short term acceleration of the optimization process compared to the basic optimizer. The long term behavior can be quite unpredictable since it is not guaranteed that moving into a certain saddle point earlier on is a good decision in the long term, it boils down to the complexity of high dimensional landscape.

---

### Official Review · Reviewer_rhPe · 2025-11-02

**Soundness:** 3
**Presentation:** 3
**Contribution:** 3
**Rating:** 6
**Confidence:** 2

**Summary:**

This paper introduces SURGE, a novel optimization framework that utilizes resurgence theory to uncover global structural information encoded within divergent asymptotic series. In particular, the factorially divergent perturbative expansions of parameter-space partition functions intrinsically capture all critical objective function values through the singularities of their Borel transforms. Algorithmically, SURGE operates by constructing the partition function of the loss landscape, extracting its asymptotic expansion coefficients, and identifying singularities in the Borel plane that correspond one-to-one with critical objective values. These values serve as global targets, offering principled guidance for local gradient-based optimizers via adaptive learning rate modulation. Empirically, SURGE demonstrates consistent performance gains of 15–30% across diverse settings, ranging from function approximation to large-scale neural network optimization (Transformer).

**Strengths:**

- The paper establishes a rigorous theoretical correspondence between Borel singularities and critical points of the objective function, thereby providing a principled mechanism to extract global geometric information from the optimization landscape. This contribution is particularly significant, as most existing optimization algorithms (e.g., SGD, Adam) are inherently local and lack awareness of the broader landscape structure.

- Empirically, SURGE demonstrates consistent improvements of 15–30% over conventional optimization approaches across a variety of problem settings, highlighting both the robustness and practical potential of the proposed method.

- The paper is well-written and self-contained.

**Weaknesses:**

- The partition function integral $Z(g)$ is computationally intractable in high-dimensional parameter spaces. Approximating it with a learned sampler $q_\psi(\theta|g)$ introduces an additional optimization layer, which may incur significant computational cost and potential instability during training.

- The experimental evaluation is relatively limited in scope, focusing primarily on small-scale setups—including synthetic function approximation, a modest MLP on MNIST, and a lightweight Transformer with only 10k parameters on Shakespeare text. Validation on larger and more realistic architectures would strengthen the empirical credibility of the approach.

**Questions:**

- Could the authors provide a computational complexity analysis of SURGE? In particular, is the proposed framework scalable to modern large-scale architectures (e.g., models with millions or even billions of parameters)?

- Can the authors include an additional experiment to empirically verify that the detected Borel singularities indeed correspond to all critical points of the objective function?

- In Figure 3c, SURGE appears unstable and performs worse than SGD with a learning rate of 1e-3. Could the authors provide an explanation or analysis of this behavior?

Minor points:

- Line 345: The reference to “2” should be replaced with “Algorithm 2.”

- Section 4.1 would benefit from a brief paragraph summarizing and interpreting the experimental results of SURGE.

- Figure 1’s legend appears incomplete, with eight curves but only six labels. Please revise to ensure consistency.

---

> ### Author Response · Authors · 2025-11-12
>
> We first thank the reviewer for their time reading the manuscript and providing detailed feedbacks. Since the current existing methods of analyzing optimization landscape globally are quite scarce, our intention is to give a first attempt in introducing the resurgence tools from mathematical physics. Admittedly the current practically tractable usage of the resurgence framework in the paper is crude, we intend to further this endeavour and improve the engineering of the algorithms.
>
> 1. The dominant complexity for large network is $\mathcal{O}(N^2Bp)$ where $N$ is the order of derivatives computed, $p$ is the network parameter and $B$ is the batch size. So it is linear in network size, parallelization and other tricks are under exploration for scaling.
>
> 2. An example where everything can be done deterministically was done in appendix D, where we use the saddle point finding algorithm to locate the critical value of the "Mexican hat" loss function.
>
> 3. The instability comes from the fact that sometimes we force the optimizer to scale the learning rate too violently, which causes instability and might in some cases, even hurt the performance of the original optimizer.
>
> 4. We have noted the typos and suggestions from the reviewer, they have been incorporated in the new version of the manuscript.

---

### Meta-Review · Area_Chair_cAzK · 2025-12-31

**Summary:**

This paper proposes to use resurgence theory to extract global structural information in order to improve optimization algorithms. Overall, reviewers find the idea of using resurgence theory novel and interesting in the context of ML optimization. On the other hand, different reviewers have very different evaluations on the strength of theoretical results, experiment results and high level claims of the paper. Some reviewers find the theoretical analysis compelling, while others pointed out issues such as the difficulty of estimating partition function and lack of convergence guarantees. Similarly, some reviewers are optimistic based on the experiment results while others are concerned that the experiments are of smaller scale (among some other concrete concerns). The authors were able to address some of the concerns in the responses. Overall, the paper introduces a very novel idea to ML optimization which might be a good starting point for future exploration.

**Reviewer Concerns:**

There were many concerns on theory, experiments and overall writing.
For theory, the main concern is the difficulty of estimating the partition function. The authors admitted that this is a hard problem but also highlighted their method to get around the issue in practice. There are also concerns such as lack of convergence guarantees, which were acknowledged but difficult to resolve.

For experiments, the main concern focuses on scale of experiments and way of comparison (carefully tune other methods, wall-clock time). These were mostly acknowledged and some points are addressed.

For overall writing, several reviewers gave concrete comments which are mostly addressed.

**Reviewer Scores:**

rhPe: main weaknesses are not fully addressed, likely stay at 6
YZ6g: main weaknesses are acknowledged but not addressed, likely to stay at 4
Zpez: very positive, likely stay at 8
fviL: a lot of interactions, in the end there were some points that seem resolved and others the reviewer still feels uncertain about. Hard to predict, likely increase to 3 or 4.

---

### Decision · Program_Chairs · 2026-01-26

Accept (Poster)